# A Survey of Deep Learning-Based Low-Light Image Enhancement

**DOI:** 10.3390/s23187763

**Published:** 2023-09-08

**Authors:** Zhen Tian, Peixin Qu, Jielin Li, Yukun Sun, Guohou Li, Zheng Liang, Weidong Zhang

**Affiliations:** 1School of Information Engineering, Henan Institute of Science and Technology, Xinxiang 453003, China; tz1128@stu.hist.edu.cn (Z.T.); ljl_hist@stu.hist.edu.cn (J.L.); sunyukun@stu.hist.edu.cn (Y.S.); ligh@hist.edu.cn (G.L.); zwd@hist.edu.cn (W.Z.); 2Institute of Computer Applications, Henan Institute of Science and Technology, Xinxiang 453003, China; 3School of Internet, Anhui University, Hefei 230039, China; zliang@ahu.edu.cn

**Keywords:** low-light Images, image degradation, image enhancement, deep learning

## Abstract

Images captured under poor lighting conditions often suffer from low brightness, low contrast, color distortion, and noise. The function of low-light image enhancement is to improve the visual effect of such images for subsequent processing. Recently, deep learning has been used more and more widely in image processing with the development of artificial intelligence technology, and we provide a comprehensive review of the field of low-light image enhancement in terms of network structure, training data, and evaluation metrics. In this paper, we systematically introduce low-light image enhancement based on deep learning in four aspects. First, we introduce the related methods of low-light image enhancement based on deep learning. We then describe the low-light image quality evaluation methods, organize the low-light image dataset, and finally compare and analyze the advantages and disadvantages of the related methods and give an outlook on the future development direction.

## 1. Introduction

Due to the development of technology and the continuous improvement of photographic equipment, we have higher and higher requirements for the quality of the images we capture, but we often have difficulty obtaining suitable images because of the interference of environmental factors. Uneven lighting, low lighting, and other factors like backlighting can result in imperfect image information, diminishing the overall quality of captured images. Figure 1 shows an example of an image under suboptimal lighting conditions. Consequently, these issues can have a cascading effect on advanced tasks such as object recognition, detection, and classification. As artificial intelligence technologies continue to evolve, the associated industries are also changing, and thus the requirements for related downstream tasks are increasing. The quality of tasks completed in the image processing area [1,2,3,4,5] can greatly affect the efficiency of upstream tasks.

In daily life, we often encounter uncontrollable environmental or equipment factors that cause uneven lighting, darkness, backlighting, and blurring of captured images [6,7,8,9]. However, we have demand for high-quality images. Superior image quality is crucial for everyday scenarios and holds significant importance across various sectors [10,11,12,13,14], including intelligent transportation and vision monitoring. Therefore, quality enhancement of images has become a subject worthy of further exploration.

The enhancement of low-light images holds a significant role in image processing. This involves enhancing the visual quality of images captured in low-light conditions by adjusting the contrast and brightness levels, thus improving visibility [15,16,17]. Conventional techniques for enhancing low-light images often revolve around statistical learning, including approaches like local exposure compensation algorithms. While these methods can enhance image brightness effectively, they may concurrently introduce undesired noise and distortion. In addition, traditional methods have some limitations [18,19,20]. While the favorable notion of adopting the reflection component as the enhancement outcome may hold inconsistently, particularly when factoring in diverse lighting attributes, it could potentially result in impractical improvements, such as the omission of details and distortion of color. Additionally, the model overlooks noise, allowing it to persist or amplify within the enhanced output [21,22,23,24,25].

In recent years, deep learning methods have made significant advancements in various fields, particularly in image processing tasks. Unlike traditional approaches, deep learning techniques place a stronger emphasis on capturing the spatial features in an image, allowing for better preservation of details and increased resistance to noise [26,27,28,29]. Deep learning has demonstrated remarkable achievements in various fields, including enhancing low-light images. In comparison with conventional approaches, deep learning-based solutions for improving low-light image quality have gained substantial attention due to their enhanced precision, robustness, and efficiency. The existing deep learning techniques for low-light image enhancement establish a relationship between an output image and a correspondingly enhanced input image under low-light conditions by designing a network structure. However, this approach can result in a high dependency on the training data, which limits its effectiveness to some extent.

This paper focuses on employing deep learning techniques for enhancing low-light images while offering an extensive assessment and analysis of current methods within this domain. The noteworthy contributions of this study encompass the following:

(1) We systematically classify and summarize the deep learning-based low-light image enhancement methods proposed in recent years, introduce the core ideas of the mentioned algorithms in detail, and provide an insightful analysis of the problems of and possible solutions for the existing methods.

(2) We summarize the datasets in the field of low-light image enhancement in detail, including the sources, characteristics, and application scenarios of the datasets. We also provide a comprehensive comparison of different datasets and discuss their respective advantages and disadvantages.

(3) We analyze the advantages and disadvantages of some existing methods through experimental comparisons, present possible problems, and look forward to future research directions.

## 2. Low-Light Image Enhancement Method Based on Deep Learning

Deep learning has gained widespread popularity and is extensively utilized in various vision processing tasks due to its efficiency and convenience. In recent years, it has emerged as a prominent topic in machine learning and has found significant applications in the field of image enhancement. Deep learning-based approaches have demonstrated exceptional performance in enhancing low-light images, making them a prominent trend in current image processing research. The purpose of low-light image enhancement is to improve the visibility and quality of images captured in low-light conditions. Nowadays, deep convolutional neural networks (CNNs) and generative adversarial networks (GANs) have been applied in several directions and have proven to be effective solutions in low-light image enhancement. In Table 1, we summarize the basic characteristics of representative methods based on deep learning. These methods utilize the power of deep learning to enhance image quality and improve visibility to provide impressive results.

### 2.1. CNN-Based Methods

Utilized for low-light image enhancement, the convolutional neural network (CNN) functions as a supervised learning approach. It effectively addresses the challenge of improving low-light images by acquiring the mapping correlation between the input and output images. This approach involves utilizing a multi-layer convolutional network to obtain higher-quality enhanced images. The standard CNN architecture consists of three key elements: the convolutional layer, the pooling layer, and the fully connected layer. These components play distinct roles, with the convolutional layer focused on extracting local features from the input image. Subsequently, the pooling layer plays a role in parameter reduction, enhancing network efficiency. Ultimately, the fully connected layer generates the desired output outcomes. By integrating these layers, a CNN effectively enhances low-light images and achieves outstanding results.

#### 2.1.1. Physical Model-Based Methods

Lore et al. [30] introduced the pioneering Low Light Net (LLNet), a deep learning-based technique for enhancing low-light images. They employed a densely layered sparse denoising autoencoder to execute contrast enhancement and denoising procedures. This groundbreaking research set the foundation for comprehensive web applications in low-light image enhancement (LLIE). Building upon this foundation, Lv et al. [33] introduced the MBLLEN, an innovative end-to-end multi-branch enhancement network. This architecture markedly improves low-light image enhancement performance, achieving this through the extraction of impactful feature representations across a feature extraction module, enhancement module, and fusion module. Furthermore, Ren et al. [56] developed a sophisticated end-to-end architecture involving an encoder-decoder network dedicated to enhancing image content alongside a recurrent neural network specifically for improving image edge enhancements. Their approach offers comprehensive enhancement capabilities, addressing both image content and edges. These studies highlight the continuous advancements in low-light image enhancement through deep learning techniques, demonstrating the potential for improving image quality in challenging lighting conditions. Tao et al. [57] introduced a learning framework based on a low-light CNN (LLCNN) to address the gradient disappearance problem in low-light image enhancement. They utilized multi-scale feature maps to mitigate this issue and incorporated SSIM loss to preserve the image texture during model training. This approach enables the adaptive enhancement of low-light images by effectively enhancing contrast while maintaining image details. Similarly, Xu et al. [58] introduced a decomposition enhancement network. Their method is centered on restoring image content by addressing noise within the low-frequency layer while highlighting intricate details in the high-frequency layer. By leveraging this approach, they achieved enhanced image quality while effectively preserving the image details. Zhu et al. [41] introduced the Edge-Enhanced Multi-Exposure Fusion Network (EEMEFN) [42], a method structured into two primary stages: multi-exposure fusion (MEF) and edge enhancement. The multi-exposure fusion network is tailored to enhancing low-light images and calculates a transfer function in both branches to produce a pair of improved images. These images are then fused using a simple averaging scheme and further refined using refinement units, resulting in improved enhancement results. These works highlight various innovative approaches in low-light image enhancement, showcasing the ongoing advancements in the field. Figure 2 provides a flow chart of CNN method combined with physical model. These methods employ different strategies, such as multi-scale feature maps, frequency-based decomposition, and fusion techniques, to achieve superior results in enhancing low-light images.

#### 2.1.2. Non-Physical Model-Based Methods

Gharbi et al. [59] introduced a novel real-time image enhancement technique that seamlessly integrates deep learning and the bilateral filter. This inventive approach reimagines the conventional bilateral filter as a deep neural network, facilitating end-to-end learning. This methodology enables real-time image enhancement but also harnesses the capabilities of deep learning for enhanced performance. Similarly, Shen et al. [60] introduced an innovative approach known as the multi-scale Retinex network (MSR-net), which fuses CNN technology with Retinex theory. This technique parallels the concept of feedforward convolutional neural networks, using diverse Gaussian convolutional kernels to create a multi-scale Retinex framework. By directly learning the mapping between low-light images and standard-brightness images, this method effectively enhances low-light images. These studies showcase innovative methods that combine deep learning techniques with well-established image enhancement theories. By integrating deep neural networks and Retinex theory, researchers have made significant strides in developing end-to-end networks that effectively enhance low-light images. Wei et al. [32] introduced a deep neural network that draws on Retinex theory to enhance images taken in low-light settings. Utilizing the principles of Retinex theory rooted in physics, this technique involves image decomposition into reflection, intermediate reflection, and shadow components, followed by individualized enhancement of each constituent element. In a similar vein, Li et al. [31] introduced LightenNet, a trainable convolutional neural network designed specifically for enhancing low-light images. Through training, the network is fed dimly lit images as the input and produces corresponding light maps as the output. The output image is then processed by a Retinex-based model to obtain the enhanced image. Although this method can achieve desirable enhancement results for some images, it still performs poorly in some challenging real-life scenes. It is worth noting that while the proposed methods show promise in improving image enhancement under low-light conditions, there is still room for further improvement, particularly in challenging real-world scenes. Cai et al. [34] introduced a method for learning a contrast enhancer for individual images using multiple exposure images. Their approach employs a deep neural network to learn the mapping from low-contrast images to high-contrast images. By leveraging the information from multiple exposure images, this method enables effective enhancement of contrast in individual images. In a similar vein, Wu et al. [61] proposed a rapid end-to-end trainable image enhancement method based on the guided filter. This approach performs enhancement while preserving high-frequency information by incorporating the concept of the guided filter. By leveraging this technique, this method achieves efficient and effective image enhancement. Additionally, Wei et al. [62] introduced an image denoising and enhancement method based on recurrent neural networks (RNNs). Their approach incorporates the concept of nonlocal mean filtering by integrating nonlocal mean filters into the RNN architecture. By utilizing nonlocal mean filters to model the spatial information of images, this method achieves effective denoising and enhancement of images. These methods highlight different approaches to image enhancement, including learning-based contrast enhancement, guided filter-based enhancement, and RNN-based denoising and enhancement. Figure 3 provides a flow chart of CNN method for non-physical model. By leveraging various techniques and models, researchers are continuously advancing the field of image enhancement, addressing challenges related to contrast, noise, and preserving image details.

Overall, the advantages of CNN-based low-light image enhancement methods lie first in the fact that local feature learning is more robust, and more image information can be extracted. Secondly, no additional processing steps are required, as it is usually an end-to-end process. In addition, the structure of CNNs is controllable and can be adapted according to the needs. Lastly, as CNNs are driven by a large amount of data, it is possible to be able to learn from the data how to perform low-light image enhancement.

Although there are many advantages to CNN-based low-light image enhancement methods, some disadvantages that cannot be hidden also need to be overcome. The first is that the global information of the image may be ignored, resulting in enhanced images that are not natural enough globally. The second disadvantage is that if the training data are insufficient, this may lead to overfitting. Finally, the performance of the CNN receives the influence of a variety of factors, which need to be debugged in a comprehensive manner.

### 2.2. GAN-Based Methods

Generative adversarial networks (GANs) constitute a class of unsupervised deep learning models composed of two primary components: a generator and a discriminator. GANs operate on the foundational concept of a competitive interplay between the generator and discriminator, fostering mutual learning to produce high-quality outcomes. The generator’s role involves creating simulated realistic samples aimed at deceiving the discriminator, which in turn specializes in discerning real samples from synthetically generated ones. In the context of low-light image enhancement, the generator focuses on extracting enhanced features from low-light images, while the discriminator evaluates the quality and authenticity of the generated images. During an iterative procedure, the generator strives to enhance its capability to generate images that closely resemble authentic ones. Meanwhile, the discriminator hones its proficiency in discerning between actual and synthetic samples. The objective is pursued until the discriminator reaches a point where it can no longer differentiate between real and generated samples. Figure 4 provides a flow chart of GAN-based method. Compared with other generative models, GANs offer the advantage of generating clear and realistic samples without relying on complex Markov chains. GANs utilize back propagation as the primary mechanism for learning, allowing for efficient training and the generation of high-quality samples.

#### 2.2.1. Condition-Based Methods

Jiang et al. [36] introduced EnlightenGAN, a GAN-based method that addresses the issue of overfitting and the limited generalization ability when training deep models on paired data. EnlightenGAN employs a U-Net architecture for generation and a composite global-local discriminator. This amalgamation involves global and local adversarial losses accompanied by self-feature preservation losses, ensuring fidelity between the augmented and authentic images. The objective behind these loss functions is to preserve coherence in the image content prior to and after enhancement, a vital aspect for maintaining structural stability during training. By combining these elements, EnlightenGAN achieves stable training and enhances the quality of low-light images. The attention-guided U-Net generator, along with the global-local discriminator and self-feature retention losses, contribute to generating more realistic and visually pleasing enhanced results, mitigating the limitations associated with overfitting and improving the generalization ability of the model. Meng et al. [63] introduced a GAN-based nighttime image enhancement framework that utilizes the properties of the GAN to generate pseudo-real images from real image distributions. The results prove its effectiveness and signify that a GAN applied to nighttime image enhancement is viable. In 2017, Ignatov et al. [64] introduced an image enhancement technique utilizing the GAN model. While this approach enhances image quality, its broader applicability is constrained by the pronounced correspondence-matching association between the initial and improved images, resulting in a heavily supervised procedure. To address this limitation, the authors later introduced an improvement using a weakly supervised network model called WESPE [65], reducing the reliance on strong supervision and yielding a more generalized algorithm. The VGG19 network calculates the content loss to maintain image content consistency, mitigating the risk of an excessively prominent correspondence match between the original and enhanced images. Furthermore, Chen et al. [66] proposed an image enhancement method based on adversarial generative networks (GANs). This approach utilizes a generator and a discriminator to achieve the mapping from ordinary photos to high-quality photos, enabling effective image enhancement. These studies demonstrate the utilization of GAN-based models for image enhancement, particularly for nighttime images. They address challenges such as strong supervision, generalizability, and quality improvement, showcasing the potential of GANs in advancing image enhancement techniques.

#### 2.2.2. Circular Consistency-Based Methods

Zhu et al. [67] introduced a generative adversarial network (GAN) framework that utilizes a recurrent consistency loss function for unpaired image transformation. This method enables learning to transform images from one domain to another while preserving the semantic information of the images. In a similar vein, Fu et al. [68] introduced LE-GAN, an innovative unsupervised low-light image enhancement network founded on generative adversarial networks. This network is trained using disparate pairs of low-light and normal-light images. To improve the quality of vision and address issues such as noise and color bias, they added an illumination-aware attention module to improve feature extraction. Additionally, a new invariant loss is introduced to tackle overexposure problems, allowing the network to adaptively enhance low-light images. These methods highlight the use of GANs for unpaired image transformation and low-light image enhancement. By incorporating recurrent consistency loss and attention modules, researchers have made significant advancements in preserving semantic information, reducing noise and color bias, and improving the visual quality of transformed and enhanced images. Yan et al. [69] introduced a low-light image enhancement method that leverages an optimization-enhanced enhancement network module within the generative adversarial network (GAN) framework. This method utilizes an enhancement network to input images into a generator, generating similar images in a new space. Subsequently, a loss function is constructed and minimized to train a discriminator, which then compares the generated images with real images to enhance the network’s performance. Similarly, You et al. [70] introduced Cycle-CBAM, a retinal image enhancement technique built upon the foundation of Cycle-GAN. This method aims to elevate the quality of fundus images from lower to higher levels without necessitating paired training data. To tackle challenges posed by texture information loss and detail degradation due to unpaired image training, Cycle-GAN is augmented by the integration of Convolutional Block Attention Module (CBAM). These strategies highlight the utilization of GAN-based methodologies in enhancing both low-light and retinal images. By optimizing the enhancement network module and incorporating attention mechanisms, researchers strive to enhance the quality and fidelity of the resulting enhanced images.

Overall, the advantages of GAN-based low-light image enhancement methods lie first in the fact that they are good at generating more realistic enhanced images. Secondly, they can learn more advanced image features, which can be used to recover image details and information. In addition, GANs are able to perform complex nonlinear modeling and capture complex features. Lastly, GANs are able to generate diversity images, which makes the results more diverse.

Although there are many advantages of GAN-based low-light image enhancement methods, there are some significant drawbacks for the direction of our future work. The first is that the training process of a GAN is complex, and the problem of unstable training may occur. The second is that the training may suffer from the problem of pattern collapse, which may lead to a lack of diversity in the results. In addition, a GAN also requires a certain amount of data to support the training of the model. Finally, the performance of a GAN receives the influence of various hyperparameters, which need to be carefully adjusted to keep it stable.

## 3. Low-Light Image Quality Evaluation

Commonly used image quality assessment (IQA) methods can be categorized into two main categories: reference-based and referenceless. Reference-based algorithms rely on having both the original (considered to be of high quality) and distorted images in order to calculate quality scores. These algorithms are commonly employed to measure the quality of images after undergoing processes such as image compression, image transfer, or image stitching. Referenceless methods, on the other hand, do not require a reference image. They estimate image quality by analyzing the characteristics and features of the distorted image itself. These methods are particularly useful when a reference image is not available or when evaluating image quality in real-time scenarios. By categorizing IQA methods into reference-based and referenceless approaches, researchers have been able to develop techniques suitable for various scenarios, enhancing our ability to assess and quantify image quality accurately.

### 3.1. Full-Reference Metrics

Commonly used reference-based image quality evaluation methods include the peak signal-to-noise ratio (PSNR) and structural similarity (SSIM) [71]. These methods evaluate image quality by comparing the target image with the original image. These evaluation methods have found widespread application in tasks such as image compression and evaluating image quality after processes like image transmission and stitching. However, it is important to note that if the reference image itself has inherently poor quality, then the credibility of these evaluation indices is significantly compromised. The accuracy of the evaluation depends on the assumption that the reference image represents a high-definition image. On the other hand, due to the complexity of the signal perceived by the human eye, different types of images, such as those with varying texture complexity or different image attributes, may yield similar PSNR or SSIM scores but can be judged differently by the human eye. This highlights the limitations of relying solely on objective evaluation metrics, as subjective human perception may vary even when objective metrics suggest similar quality. Overall, while the PSNR and SSIM serve as widely used reference-based evaluation metrics, the credibility of their results depends on the quality of the reference image, and the subjective perception of image quality can differ based on various image attributes and human interpretation.

In 2004, a research paper published by the University of Texas at Austin presented SSIM, a metric employed to gauge the similarity between two images. The metric assigns a value between 0 and 1, where higher values signify increased image similarity and lower values indicate greater dissimilarity. SSIM is a comprehensive reference tool to assess the likeness of x and y images. The calculation of SSIM is as follows:$$l\left(m,n\right)=\frac{2\mu_{m}\mu_{n}+C_{1}}{\mu_{m}^{2}+\mu_{n}^{2}+C_{1}},$$
(1)$$c\left(m,n\right)=\frac{2\sigma_{m}\sigma_{n}+C_{2}}{\sigma_{m}^{2}+\sigma_{n}^{2}+C_{2}},$$
$$s\left(m,n\right)=\frac{\sigma_{mn}+C_{3}}{\sigma_{m}\sigma_{n}+C_{3}}.$$

In the above equation, l(m,n) is the mean to estimate the brightness, c(m,n) is the variance to estimate the contrast, and s(m,n) is the covariance to estimate the structural similarity. Meanwhile, $\mu_{m}$ and $\mu_{n}$ represent the mean of *m* and *n*, respectively, $\sigma_{m}$ and $\sigma_{n}$ represent the standard deviation of *m* and *n*, respectively, $\sigma_{mn}$ represents the covariance of *m* and *n*, and $C_{1}$, $C_{2}$, and $C_{3}$ all represent constants. Therefore, SSIM is defined as
(2)$$\mathrm{SSIM}\left(m,n\right)=\frac{\left(2\mu_{m}\mu_{n}+C_{1}\right)\left(2\sigma_{mn}+C_{2}\right)}{\left(\mu_{m}^{2}+\mu_{n}^{2}+C_{1}\right)\left(\sigma_{m}^{2}+\sigma_{n}^{2}+C_{2}\right)},$$
where SSIM takes a value between 0 and 1. When two images are identical, the SSIM value is one.

The PSNR is a widely used objective criterion for evaluating images in engineering projects. It measures the ratio between the maximum signal and the background noise and is commonly employed to assess the amount of information loss in a compressed image compared with the original. The PSNR is measured in decibels (dB), with higher values indicating better image quality (i.e., less noise). The range of PSNR values is from 0 to positive infinity.

The PSNR is often calculated using the mean squared error (MSE). Considering a pair of monochrome images labeled as *I* and *J*, where *I* signifies an unadulterated original image and *J* denotes an angry rendition of *I* (e.g., *I* as the uncompressed source image and *J* as its compressed iteration), the calculation of their mean squared error involves the following:(3)$$\mathrm{MSE}=\frac{1}{xy}\sum_{i=1}^{x}\sum_{j=1}^{y}{\left(I\left(i,j\right)-J\left(i,j\right)\right)}^{2}.$$

The PSNR is defined as
(4)$$\mathrm{PSNR}=10\cdot \log_{10}\left(\frac{{\mathrm{MAX}}_{I}^{2}}{\mathrm{MSE}}\right)=20\cdot \log_{10}\left(\frac{{\mathrm{MAX}}_{I}}{\sqrt{\mathrm{MSE}}}\right),$$
where MAX is the maximum pixel value of the image and the PSNR is measured in decibels (dB). A higher PSNR value indicates better image quality. Generally speaking, PSNR values above 40 dB are considered excellent, meaning the image quality is very close to the original. PSNR values between 30 dB and 40 dB typically indicate good image quality, with detectable but acceptable distortion. On the other hand, PSNR values between 20 dB and 30 dB signify poor image quality. PSNR values below 20 dB indicate unacceptable image quality. It is worth noting that PSNR values below −30 dB also indicate poor image quality and are considered unacceptable.

### 3.2. Non-Reference Metrics

In evaluating non-reference images, image clarity is a crucial indicator for measuring image quality. It closely aligns with the subjective perception of individuals, as low image clarity indicates a blurred image. Traditional reference-free methods for image quality evaluation, such as the Brenner gradient function, Tenengrad gradient function, and image information entropy function (Information entropy), can provide insights into the level of image clarity to a certain extent. These methods contribute to the assessment of image quality by quantifying the clarity of the image, thereby facilitating objective evaluations of image quality in the absence of reference images.

The Brenner gradient function serves as a fundamental gradient assessment approach involving the computation of the squared difference between adjacent pixel grayscale values. This function is defined as follows:(5)$$B\left(f\right)=\sum_{n}^{}\sum_{m}^{}{\left|f(m+2,n)-f(m,n)\right|}^{2},$$
where f(m,n) denotes the grayscale value of the image *f* corresponding to the pixel (m,n) and B(f) is the result of the image sharpness calculation.

The Tenengrad gradient function applies the Sobel operator to capture gradients both horizontally and vertically. The definition for image sharpness within the context of the base and Tenengrad gradient functions is outlined as follows:(6)$$B\left(f\right)=\sum_{n}^{}\sum_{m}^{}\left|E(m,n)\right|\left(E\left(m,n\right)>Q\right).$$

E(m,n) takes the following form:(7)$$E\left(m,n\right)=\sqrt{E_{m}^{2}\left(m,n\right)+E_{n}^{2}\left(m,n\right)},$$
where *Q* is the given edge detection threshold and $E_{m}$ and $E_{n}$ are the convolution of the Sobel horizontal and vertical edge detection operators at pixel point (m,n), respectively.

The information entropy function stands as a crucial metric for assessing the information abundance within an image. As established by information theory, the information held within an image *f* is quantified through the information entropy B(f) associated with said image:(8)$$B\left(f\right)=-\sum_{a=0}^{G-1}P_{a}\;\ln\left(P_{a}\right),$$
where $P_{a}$ signifies the probability of encountering a pixel with the gray value *a* within the image and *G* represents the total count of gray levels (typically set at 256). As per Shannon’s information theory, maximal information is attained when entropy reaches its peak. Applying this principle to image focusing, heightened B(f) values correlate with sharper images. It is noteworthy that the entropy function’s sensitivity is not particularly pronounced, and due to image content variations, outcomes might occasionally deviate from actual scenarios.

## 4. Benchmark Dataset

Deep learning relies on deep neural networks that require extensive training with substantial data samples to achieve generalizability in the final model. Consequently, the dataset size significantly influences deep learning endeavors. To cater to the needs of low-light image enhancement research, there are several publicly available datasets with varying sizes and diverse scenarios. Table 2 provides summary of different low-light image data. These datasets include naturalness preserved enhancement (NPE), Vasileios Vonikakis (VV), the low-light dataset (LOL), and multi-exposure image fusion (MEF), Figure 5 and Figure 6 provide examples of different datasets. Researchers can utilize these datasets to train and evaluate their low-light image enhancement models, providing a range of options to suit different research requirements.

(1)NPE dataset

In 2013, Wang et al. [72] introduced the NPE dataset, which comprises two components: a natural image dataset and a low-light image dataset. The natural image dataset consists of a diverse collection of indoor and outdoor scenes captured under normal lighting conditions. In contrast, the low-light image dataset is derived from natural images through specific processing techniques that simulate low-illumination conditions. The low-light image dataset further categorizes the images into two levels: “low” and “high”, which represent different degrees of low-light situations.

(2)MEF dataset

In 2015, Ma et al. [73] introduced the MEF dataset, comprising images capturing multiple low-light scenes. The dataset encompasses a diverse range of scenes and shooting conditions, including indoor and outdoor environments, various lighting scenarios, candlelight, and daytime and nighttime shots. It has found extensive usage in both academic and industrial settings, serving a wide array of applications. These include the research and development of image enhancement algorithms, testing, and evaluation of digital cameras, among other related areas. The MEF dataset has proven to be a valuable resource for advancing image processing techniques, enabling advancements in the field of low-light photography, and supporting various imaging technology-related endeavors.

(3)VV dataset

The VV dataset was explicitly designed to explore images that are significantly underexposed or overexposed, exhibiting a key characteristic of having areas that are properly exposed alongside regions that are grossly underexposed or overexposed. This dataset serves as a valuable resource for evaluating algorithms focused on local exposure correction and enhancement. It provides an ideal testing ground for techniques aiming to address the challenges posed by varying exposure levels within an image.

(4)SID dataset

In 2018, Chen et al. [74] introduced the SID dataset to facilitate the advancement of deep learning approaches in low-light image processing. The dataset encompasses 5094 original short-exposure images, each with a matching long-exposure reference image. These image pairs were obtained using two distinct camera models: the Sony α7SII and the Fujifilm X-T2. Raw data were collected under low-light conditions with short exposure times, typically around 0.1 seconds or 0.04 seconds. The ground truth images were captured with long exposure times of 10 or 30 seconds, ensuring minimal noise presence. The SID dataset offers a valuable resource for training and evaluating deep learning models in the domain of low-light image processing, providing paired images with well-defined exposure characteristics.

(5)LOL dataset

In 2018, Wei et al. [32] introduced the LOL dataset, a paired collection comprising 500 low-light and normal-light image pairs. These pairs were further divided into 485 training pairs and 15 test pairs for evaluation purposes. The low-light images in the dataset accurately represent the noise typically encountered during the process of capturing photographs. The majority of the images depict indoor scenes and are sourced from diverse scenes and devices, including cell phones and digital cameras. As a result, the dataset encompasses a wide variety of objects captured within the images. Additionally, the LOL dataset covers various low-light conditions, including twilight and indoor low-light scenarios, providing a comprehensive representation of challenging lighting situations. To ensure consistency and comparability, all raw images were adjusted to a standardized resolution of 400 × 600 pixels and converted to the portable web graphics format. Its diverse range of images and realistic representation of low-light conditions make it a valuable tool for research and development in the field.

(6)SICE dataset

In 2019, Cai et al. [34] presented the SICE dataset, an extensive collection of multi-exposure images. The dataset creation process involved deploying multi-exposure fusion (MEF) and high dynamic range (HDR) techniques to reconstruct reference images, yielding heightened contrast and visibility improvements. To create the SICE dataset, the authors employed 1200 sequences and applied 13 MEF and HDR algorithms, resulting in a total of 15,600 fusion results (1200 sequences × 13 algorithms). The dataset comprises 589 meticulously selected high-resolution multiple-exposure sequences, consisting of a total of 4413 images. For each sequence, a set of contrast-enhanced images was produced using 13 diverse multiple-exposure image fusion techniques and a stack-based high dynamic range imaging algorithm. Following this, subjective evaluations were carried out to determine the optimal reference images for each scene.

(7)ExDark dataset

In 2019, Loh et al. [75] introduced the Exclusively Dark (ExDark) dataset, consisting of 7363 low-light images. The dataset covers a range of low-light conditions from very low-light environments to twilight, encompassing 10 different lighting conditions. The images were captured in various real-world scenarios using a diverse array of devices and cameras, providing a wide representation of different scenes and shooting conditions. These low-light images are subject to multiple factors, including insufficient lighting, noise, and blur, which further contribute to the challenges associated with low-light photography. The dataset includes annotations at both the image class level and the local object bounding box level for 12 object classes, such as bicycle, car, cat, dog, chair, and cup. The ExDark dataset serves as a valuable resource for researchers and practitioners in the field of low-light image analysis. Its comprehensive collection of low-light images, diverse lighting conditions, and detailed object annotations provide an excellent foundation for developing and evaluating algorithms related to object recognition, localization, and other tasks in challenging low-light environments.

(8)RELLISUR dataset

In 2021, Aakerberg et al. [76] introduced the RELLISUR dataset, a large-scale paired dataset specifically designed for low-light and low-resolution image enhancement tasks. The RELLISUR dataset comprises 12,750 paired images, consisting of real low-light and low-resolution images paired with high-resolution reference images captured under normal lighting conditions. The dataset covers a wide range of resolutions and low-light levels, enabling the development and training of deep learning-based models. It enables the exploration and development of deep learning models that can effectively enhance the quality and resolution of low-light images, bridging the gap between these two important image enhancement tasks.

(9)LLIV-Phone dataset

In 2021, Li et al. [77] introduced the LLIV-Phone dataset, a comprehensive and challenging dataset specifically designed for low-illumination image and video analysis. The LLIV-Phone dataset comprises 120 videos and 45,148 images captured using 18 different cell phone cameras. The dataset covers a wide range of indoor and outdoor scenes with diverse lighting conditions, including low light, underexposure, moonlight, dusk, darkness, extreme darkness, backlighting, non-uniform lighting, and colored lighting. These real-world scenes present various challenges associated with low-light conditions. It offers a wide variety of low-light images and videos collected from real scenes, making it suitable for testing and comparing the performance of different enhancement algorithms. By providing a comprehensive collection of real low-illumination images and videos, the LLIV-Phone dataset significantly contributes to the advancement of research in low-light image and video enhancement, enabling the development and evaluation of robust algorithms in this domain.

## 5. Experimental Evaluation and Analysis

### 5.1. Performance Comparison

To evaluate and analyze different methods, experiments were conducted using various datasets, including NPE [72], MEF [73], VV, and ExDark [75]. The comparison methods considered in the study included CERL [78], Zero-DCE [39], Zero-DCE++ [45], and SCI [49]. For quantitative analysis, the researchers selected the peak signal-to-noise ratio (PSNR) and structural similarity (SSIM) as evaluation metrics. These metrics were used to measure and analyze the performance of the selected methods. The researchers utilized the pretrained models provided in the original papers for validation and comparison. The comparison results were divided into two main parts: qualitative and quantitative evaluation. Qualitative evaluation involves human perception and judgment of the visual quality of the enhanced images, due to the unique capabilities and sensitivities of the human visual system in image processing and perception. The quantitative evaluation focuses on quantitative metrics, such as the PSNR and SSIM, to provide a more systematic analysis of the performance of the methods. By conducting qualitative and quantitative evaluations, the researchers aimed to comprehensively assess and compare the selected methods by using multiple criteria to elucidate their strengths and weaknesses in enhancing low-light images.

### 5.2. Qualitative Evaluation

Qualitative evaluation involves comparing the visual differences between the enhanced image and the original low-light image based on human perception. It typically entails presenting the same low-illumination image processed by different algorithms to individuals, who are then asked to evaluate and select the best algorithm based on their visual judgment. However, qualitative evaluation is susceptible to various external factors that can introduce subjectivity and make it challenging to establish a fixed standard. Factors like individual aesthetics, color preferences, and variations in observation angles can influence the evaluation. These subjective factors contribute to the difficulty of achieving a standardized and scientific subjective evaluation. While qualitative evaluation provides valuable insights into the visual effect and overall preference of the enhanced images, it is important to acknowledge its limitations and potential biases. Therefore, it is often combined with quantitative evaluation methods, such as quantitative metrics like the PSNR and SSIM, which provide more standardized and measurable criteria for assessing the performance of image enhancement algorithms.

Figure 7 presents a qualitative comparison of several different methods using four datasets. Upon examining the qualitative comparison images in Figure 7, certain observations can be made regarding the visual effects of the algorithms. The Zero-DCE [39] algorithm model demonstrated superior subjective visual effects. The enhanced image exhibited appropriate brightness without overexposure, and it effectively extracted detailed information from the image, resulting in a more visually pleasing outcome. In contrast, the CERL [78] algorithm model did not appear to be particularly effective in terms of overall image enhancement. The quantitative evaluation suggests that it may not achieve satisfactory results for improving the overall visual quality of the images. The SCI-easy [49] algorithm model showed room for improvement, particularly in addressing local darkness. The images revealed uneven enhancement in certain areas, indicating the need for further refinement to achieve more consistent results. On the other hand, the SCI-medium [49] algorithm model performed well in enhancing extremely dark images. However, for certain specific images, there was an issue of overexposure, where certain areas of the image appeared to be excessively bright. These subjective observations highlight the strengths and weaknesses of each algorithm in terms of their visual effects.

### 5.3. Quantitative Evaluation

Quantitative evaluation involves assessing image enhancement algorithms using specific criteria and quantitative measurements. In image enhancement, this evaluation method quantifies the differences between the original and enhanced images using mathematical models and calculates the indicators to determine the quality of the image. Quantitative evaluation methods are characterized by their simplicity, computational efficiency, and ability to provide quantitative assessments based on established models, ensuring high stability and reproducibility. The evaluation indicators used in the compared methods in this paper include the PSNR and SSIM. These quantitative evaluation metrics enable a quantitative assessment of image enhancement algorithms, allowing for direct comparisons and providing a basis for performance analysis.

Table 3, Table 4 and Table 5 provide a quantitative analysis of several different methods, using the PSNR, SSIM, and NIQE as evaluation metrics. In these three metrics, the higher the values of the PSNR and SSIM, the better the result, reflecting an improvement in image quality, while the opposite is true for the NIQE; the smaller the value of the NIQE, the better the perceived quality. Upon reviewing the tables, it can be observed that the SCI-easy [49] algorithm consistently achieved excellent results for the PSNR, SSIM, and NIQE. The algorithm’s performance surpassed that of the other methods evaluated, demonstrating better quantitative metrics in terms of image quality. These quantitative assessments provide valuable insights into the algorithms’ performance and their ability to enhance image quality. It is important to consider these metrics along with qualitative evaluations and other factors to gain a comprehensive understanding of the algorithms’ effectiveness in different aspects.

## 6. Summary and Outlook

Low-light image enhancement plays a crucial role in various applications, both in everyday life and in industries. Deep learning technology has emerged as a powerful tool for tackling this challenge. However, there are still certain limitations associated with deep learning-based approaches. One notable drawback is the heavy reliance on a substantial amount of training data, which can be time-consuming to collect and annotate. Moreover, the selection of the dataset needs to be carefully considered to ensure it represents a wide range of scenes and categories, which adds complexity to the training process. Additionally, some existing deep learning methods for low-light image enhancement primarily focus on improving model performance, potentially overlooking the practicality and generalizability of the results. It is crucial to address these limitations and propose algorithms that benefit higher-level image processing tasks. Efforts should be made to develop techniques that require less training data and have shorter training times while still achieving high-quality results. Furthermore, it is important to prioritize the practical application and universality of the enhanced images. By addressing these challenges, researchers can pave the way for more efficient and effective low-light image enhancement methods that align with the needs of various image processing tasks in real-world scenarios.

Finally, this paper will introduce some possible future research directions. First of all, integrating models from different tasks to achieve multi-task image enhancement is an important area for exploration. By combining the strengths of various models, researchers can develop more comprehensive and versatile solutions. Secondly, addressing the generalization problem of deep learning models is crucial. Enhancing the models’ ability to perform well in diverse scenarios and adapt to different image characteristics will be a key focus. This can involve techniques such as transfer learning, domain adaptation, or developing robust feature representations. Lastly, improving the speed and efficiency of deep learning models is a prominent concern. Developing algorithms that can achieve real-time or near-real-time processing for low-light image enhancement tasks is essential for practical applications.

In summary, deep learning-based low-light image enhancement techniques hold great potential in computer vision and image processing domains. Future research should focus on exploring more efficient and accurate deep-learning models to achieve superior image enhancement effects. By addressing these research directions, we can advance the state of the art in low-light image enhancement and facilitate its broader applications in various fields.

## Figures and Tables

**Figure 1 sensors-23-07763-f001:**
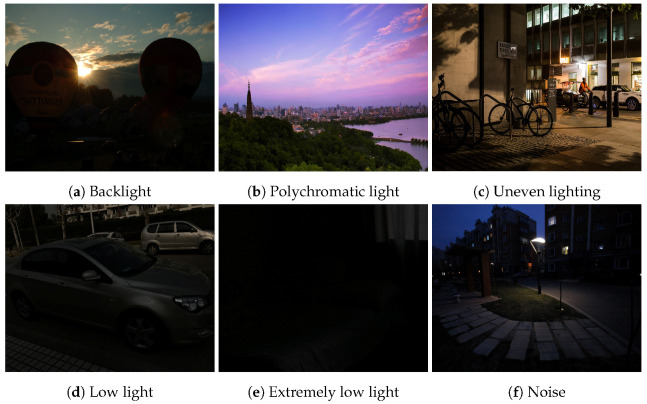
Examples of images under suboptimal lighting conditions.

**Figure 2 sensors-23-07763-f002:**
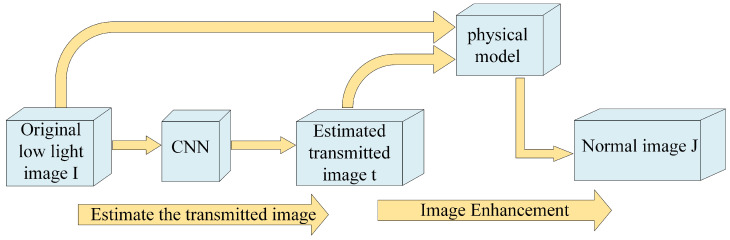
Flow chart of CNN method combined with physical model.

**Figure 3 sensors-23-07763-f003:**
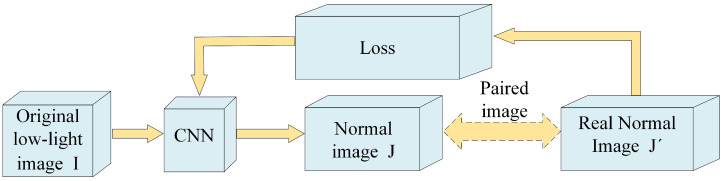
Flow chart of CNN method for non-physical model.

**Figure 4 sensors-23-07763-f004:**
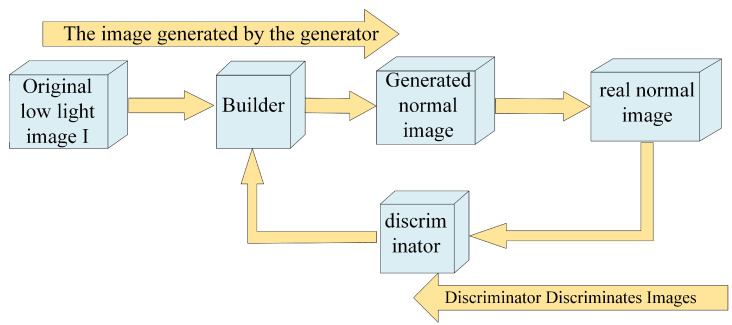
Flow chart of GAN-based method.

**Figure 5 sensors-23-07763-f005:**
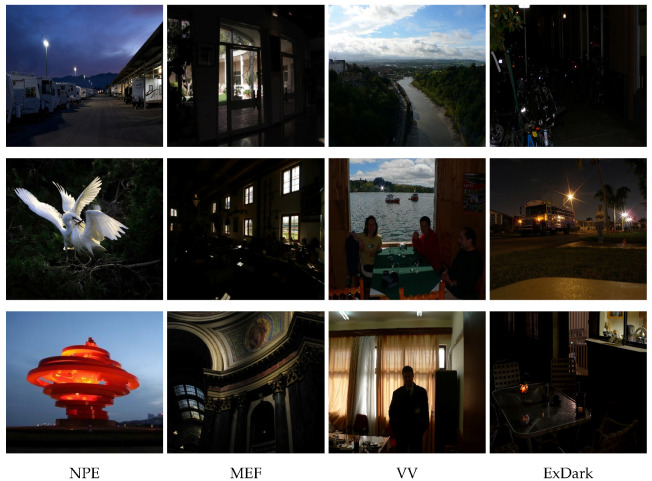
Example of a partial low-light dataset.

**Figure 6 sensors-23-07763-f006:**
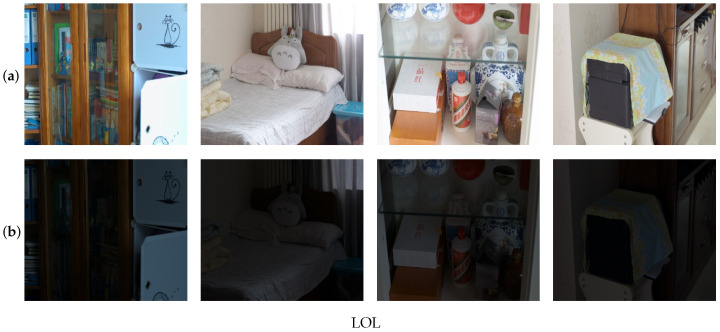
Example of paired low-light dataset for LOL. Here, (**a**) is a reference image, and (**b**) is a low light image.

**Figure 7 sensors-23-07763-f007:**
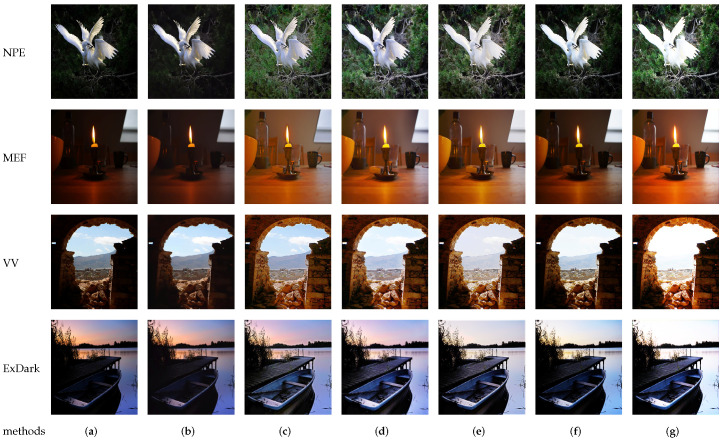
Qualitative comparison of different low-light image enhancement algorithms. Where, (**a**) represents the original image, (**b**) represents the result map after processing by the CERL [78] method, (**c**) represents the result map after processing by the Zero-DCE [39] method, (**d**) represents the result map after processing by the Zero-DCE++ [45] method, (**e**) represents the result map after processing by the SCI-difficult [49] method, (**f**) represents the result map after processing by the SCI-easy [49] method, and (**g**) represents the results of the SCI-medium [49] method.

**Table 1 sensors-23-07763-t001:** Summary of the basic characteristics of representative methods based on deep learning.

Year	Methods	Network Structure	Training Data	Test Data	Evaluation Metric	Platform
2017	LLNet [30]	SSDA	Simulated by gamma correction and Gaussian noise	Simulated self-selected	PSNR SSIM	Theano
2018	LightenNet [31]	Four layers	Simulated by random illumination values	Simulated self-selected	PSNR MAE SSIM user study	Caffe MATLAB
	Retinex-Net [32]	Multi-scale network	LOL simulated by adjusting histogram	Self-selected	-	TensorFlow
	MBLLEN [33]	Multi-branch fusion	Simulated by gamma correction and Poisson noise	Simulated self-selected	PSNR SSIM AB VIF LOE TOMI	TensorFlow
	SICE [34]	Frequency decomposition	SICE	SICE	PSNR FSIM runtime FLOPs	Caffe MATLAB
2019	KinD [35]	Three subnetworks U-Net	LOL	LOL LIME NPE MEF	PSNR SSIM LOE NIQE	TensorFlow
	EnlightenGAN [36]	U-Net-like network	Unpaired real images	NPE LIME MEF DICM VV BBD-100K ExDARK	User study NIQE classification	PyTorch
	ExCNet [37]	Fully connected layers	Real images	IEpxD	User study CDIQA LOD	PyTorch
	DeepUPE [38]	Illumination map	Retouched image pairs	MIT-Adobe FiveK	PSNR SSIM user study	TensorFlow
2020	Zero-DCE [39]	U-Net-like network	SICE	SICE NPE LIME MEF DICM VV DARK FACE	User study PI PNSR SSIM MAE runtime face detection	PyTorch
	DRBN [40]	Recursive network	LOL images selected by MOS	LOL	PSNR SSIM SSIM-GC	PyTorch
	EEMEFN [41]	U-Net-like network edge detection network	SID	SID	PSNR SSIM	TensorFlow Paddle
	TBEFN [42]	Three stages U-Net-like network	SCIE LOL	SCIE LOL DICM MEF NPE VV	PSNR SSIM NIQE runtime P FLOPs	TensorFlow
	DSLR [43]	Laplacian pyramid U-Net-like network	MIT-Adobe FiveK	MIT-Adobe FiveK self-selected	PSNR SSIM NIQMC NIQE BTMQI CaHDC	PyTorch
2021	RUAS [44]	Neural architecture search	LOL MIT-Adobe FiveK	LOL MIT-Adobe FiveK	PSNR SSIM runtime P FLOPs	PyTorch
	Zero-DCE++ [45]	U-Net-like network	SICE	SICE NPE LIME MEF DICM VV DARK FACE	User study PI PNSR SSIM P MAE runtime face detection FLOPs	PyTorch
	DRBN [46]	Recursive network	LOL	LOL	PSNR SSIM SSIM-GC	PyTorch
	RetinexDIP [47]	Encoder-decoder networks	-	DICM, ExDark fusion LIME NASA NPE VV	NIQE NIQMC CPCQI	PyTorch
	PRIEN [48]	Recursive network	MEF LOL simulated by adjusting histogram	LOL LIME NPE MEF VV	PNSR SSIM LOE TMQI	PyTorch
2022	SCI [49]	Self-calibrated illumination network	MIT LOL LSRW DARK FACE	MIT LSRW DARK FACE ACDC	PSNR SSIM DE EME LOE NIQE	PyTorch
	LEDNet [15]	Encoder-decoder networks	LOL-Blur	LOL-Blur	PSNR SSIM MUSIQ NRQM NIQE	PyTorch
	REENet [50]	Three subnetworks	SID	SID	PSNR SSIM VIF NIQE LPIPS	TensorFlow
	LANNet [51]	U-Net-like network	LOL SID	LOL SID	PSNR SSIM GMSD NLPD NIQE DISTS	PyTorch
2023	LPDM [52]	Diffusion model	LOL	LIME DICM MEF NPE	SSIM PSNR MAE LPIPS NIQE BRISQUE SPAQ	PyTorch
	FLW-Net [53]	Two-stage network	LOL-V1 LOL-V2	LOL-V1 LOL-V2	PSNR SSIM NIQE	PyTorch
	NeRCo [54]	Encoder-decoder networks	LSRW	LOL LSRW LIME	PSNR SSIM NIQE LOE	PyTorch
	SKF [55]	Encoder-decoder networks	-	LOL LOL-v2 MEF LIME NPE DICM	PSNR SSIM LPIPS NIQE	PyTorch

**Table 2 sensors-23-07763-t002:** Summary of different low-light image datasets.

Name	Year	Quantity	Features	Type
NPE	2013	84	Multi-scene natural images	Real
MEF	2015	17	Fusion images	Real
VV	/	24	Uneven local exposure	Real
SID	2018	5094	Combination of long and short exposures	Real
LOL	2018	500 pairs	Paired normal and low-light images	Synthetic + Real
SCIE	2019	4413	Large-scale multi-exposure images	Real
ExDark	2019	7363	Multi-category, multi-scene	Real
RELLISUR	2021	12,750	Different resolutions, pairs	Real
LLIV-Phone	2021	45,148	Large scale, image and video	Real

**Table 3 sensors-23-07763-t003:** Quantitative comparison of different deep learning low-light image enhancement algorithms in terms of PSNR.

Metrics	PSNR					
Methods	CERL	Zero-DCE	Zero-DCE++	SCI-difficult	SCI-easy	SCI-medium
NPE	17.932	14.509	13.963	13.989	18.892	12.185
MEF	17.537	11.8	11.841	11.842	18.693	10.279
VV	18.006	15.606	13.984	14.016	17.459	11.481
ExDark	17.188	15.468	12.62	12.633	14.35	9.898

**Table 4 sensors-23-07763-t004:** Quantitative comparison of different deep learning low-light image enhancement algorithms in terms of SSIM.

Metrics	SSIM					
Methods	CERL	Zero-DCE	Zero-DCE++	SCI-difficult	SCI-easy	SCI-medium
NPE	0.729	0.322	0.323	0.326	0.677	0.268
MEF	0.768	0.428	0.438	0.432	0.791	0.417
VV	0.689	0.500	0.532	0.528	0.790	0.488
ExDark	0.702	0.567	0.580	0.583	0.797	0.535

**Table 5 sensors-23-07763-t005:** Quantitative comparison of different deep learning low-light image enhancement algorithms in terms of NIQE.

Metrics	NIQE						
Methods	Input	CERL	Zero-DCE	Zero-DCE++	SCI-difficult	SCI-easy	SCI-medium
NPE	4.319	2.959	3.082	2.552	3.082	2.838	3.052
MEF	4.265	3.760	3.156	3.434	3.963	3.261	3.201
VV	3.525	2.615	3.145	3.211	2.815	2.740	3.083
ExDark	4.435	3.609	3.651	2.729	3.836	3.284	3.882

## Data Availability

The data presented in this study are publicly available data (sources stated in the citations). Please contact the corresponding author regarding data availability.

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
