# Peer review of "A Survey of Deep Learning-Based Low-Light Image Enhancement"

_sensors, 2023, doi:10.3390/s23187763_

Round 1

Reviewer 1 Report

The goal of the presented research is to provide a comprehensive review of deep learning-based methods for low-light image enhancement. The main achievements are systematically classifying and summarizing recent algorithms, analyzing datasets, evaluation metrics, and comparing different methods through experiments. The authors provide insights into the state-of-the-art and future directions in this field.

There do not appear to be any major methodical or scientific issues.
The literature review and experimental methodology seem sound.

Yes, the references are consistently listed in IEEE format.

No other major problems noted. The overall structure and content flow logically.

The writing is clear, there are only a few minor issues, e.g.:

line 393: 15 test pairs are not enough, however, it is probably easy to address this issue.

line 468-470: "Qualitative evaluation involves human perception and judgment of the visual quality of the enhanced images." - Please give more detail.

Overall, the paper covers the topic comprehensively, has solid methodology, and is well-written.
Some minor edits could further polish it for publication.

Reference 4: Inconsistency in author name listing format

Reference 12: "nderwater" should be "Underwater".

multiple references: The journal title "IEEE Transactions on Image Processing" are written with lower case incorrectly

Reference 18: There are incorrect spaces between "L 2 L p".

Reference 20: Incorrect formatting of author name listing

multiple references: "arXiv preprint arXiv:...". - It might be redundant to use "arXiv preprint" before the identifier.

Reference 11: "1-1"  should be reviewed and corrected.

There are a few minor grammatical or typo issues:

line 26: "urgent" - unnecessary word here, consider changing it to "high"

line 328: missing space before "takes"

Reference 5: ".In" - missing space

References 5 and 29: "Proceedings of the Proceedings of the IEEE/CVF conference on computer vision and pattern recognition" - typo

Reviewer 2 Report

In this paper, a series of analysis and summary are made for the low-light image enhancement method based on deep learning, which is very meaningful, but there are still some shortcomings:

1. In Table 1, in the column of the methods, some add the year in front of the method, and some do not add it. It might be better to write the year of the method in a separate column.

2. In Section 2, the author only analyzes the methods based on CNN and GANs, and does not analyze the Transformer method which is a hot topic in recent years.

3. The author described some methods based on CNN and GANs in sections 2.1.1, 2.1.2, 2.2.1 and 2.2.2 in recent years, but the analysis of advantages and disadvantages is not specific and detailed enough.

4. It can be seen from the author's summary in Table 1 that there are a variety of evaluation metrics for low-light image enhancement, but few are summarized in Section 3.

5. In the quantitative analysis of the experimental section, the author only analyzed the full reference metrics, and did not analyze the performance of the no-reference metrics.

Minor editing of English language required.

Round 2

Reviewer 2 Report

The author's reponse basically solved my issue and I have no other comments or suggestions.